# On Truthful Item-Acquiring Mechanisms for Reward Maximization

Submission ID: 178

## ABSTRACT

In this research, we study the problem that a collector acquires items from the owner based on the item qualities the owner declares and an independent appraiser's assessments. The owner is interested in maximizing the probability that the collector acquires the items and is the only one who knows the items' factual quality. The appraiser performs her duties with impartiality, but her assessment may be subject to random noises, so it may not accurately reflect the factual quality of the items. The main challenge lies in devising mechanisms that prompt the owner to reveal accurate information, thereby optimizing the collector's expected reward. We consider the menu size of mechanisms as a measure of their practicability and study its impact on the attainable expected reward. For the single-item setting, we design optimal mechanisms with a monotone increasing menu size. Although the reward gap between the simplest and optimal mechanisms is bounded, we show that simple mechanisms with a small menu size cannot ensure any positive fraction of the optimal reward of mechanisms with a larger menu size. For the multi-item setting, we show that an ordinal mechanism that only takes the owner's ordering of the items as input is not incentive-compatible. We then propose a set of Union mechanisms that combine single-item mechanisms. Moreover, we run experiments to examine these mechanisms' robustness against the independent appraiser's assessment accuracy and the items' acquiring rate.

## CCS CONCEPTS

• **Theory of computation → Algorithmic mechanism design**.

## KEYWORDS

Item-Acquiring Mechanism, Reward Maximization, Incentive Compatibility

**ACM Reference Format:**
Submission ID: 178. 2024. On Truthful Item-Acquiring Mechanisms for Reward Maximization. In *Proceedings of The ACM Web Conference 2024 (TheWebConf2024)*. ACM, Singapore, 11 pages. https://doi.org/XXXXXXX.XXXXXXX

## 1 INTRODUCTION

Information asymmetry is a prevalent concern in online markets and the economics of the web. This situation arises when one party possesses superior information compared to another in a market setting, potentially leading to inefficiencies that hinder the quantity and quality of transactions. To address these issues and enhance market efficiency, it becomes imperative to design incentive mechanisms that encourage truthful information sharing among the involved parties.

In one such scenario, an owner possesses a collection of items, and their actual quality is undisclosed. A collector considers acquiring these items but lacks precise information about their condition and quality. Despite this, the collector has access to statistical data related to the items, offering valuable yet potentially imperfect insights. Furthermore, the collector can rely on assessments from an independent appraiser, an expert in the field. It's important to note that these appraiser assessments may be affected by random noise, adding complexity to the decision-making process. The central question addressed in this study revolves around whether the collector can craft incentive mechanisms that effectively elicit truthful information from the owner. Our ultimate aim is to optimize the collector's reward for acquiring these items, shedding light on how to bridge information gaps and foster more efficient dynamics within online markets.

This item-acquiring problem encompasses a wide range of optimization problems. We provide several illustrative examples to showcase its versatility.

*Paper acceptance for conference proceedings.* The internet and web technologies have rapidly evolved, leading to a surge in academic paper submissions, especially in areas like Web economics, social networks, and user modeling. For instance, The Web Conference (WWW) saw a substantial increase in paper submissions, from around 950 in 2017 to over 1,900 in 2023, reflecting heightened interest in this dynamic field. To manage this influx, conferences employ strategies like engaging expert reviewers [32]. They also implement measures to streamline the process [10, 11, 34, 37, 40]. However, information asymmetry poses challenges in conducting accurate reviews, partly due to differences in authors' and reviewers' timeframes [34, 37]. In this process, authors aim to maximize their paper's acceptance chances, while conference organizers seek to uphold their prestige by accepting top-tier research papers.

*App Store Review Process.* The App Store Review Process exemplifies a web-related economic activity where developers submit their mobile applications to platforms like Apple's App Store or Google Play. In this ecosystem, expert moderators meticulously assess the apps for quality, security, and guidelines adherence, maintaining a high-quality, secure marketplace. App stores aim to ensure compliance, generate revenue, promote fairness, and build user trust. On the other hand, developers possess a significant information advantage over app stores and their moderators. They have in-depth knowledge about their app's functionality, marketing strategies, and monetization data, and have their own set of goals, including seeking exposure, monetization opportunities, user engagement, compliance with platform rules, receiving feedback, and achieving long-term success.

This paper presents a fresh perspective on eliciting true information in the item-acquiring problem and addresses it using mechanism design approaches. In mechanism design, a menu refers to a collection of options or alternatives offered to participants by a mechanism. Generally, a mechanism with a larger menu size has the potential to achieve higher efficiency by providing more choices to participants. However, this increased efficiency comes at the cost of greater complexity. As a result, simpler mechanisms with a smaller menu size are often preferred over complex ones with larger menus, as they are easier to implement and understand [4, 21, 38]. Mechanism design studies have extensively explored the tradeoff between menu size and optimality. The optimal size of a menu in a mechanism is often context-specific and contingent on the objectives of the mechanism designer. Taking into account these factors, researchers aim to strike a balance between offering ample choices and ensuring manageable implementation for more effective and efficient mechanisms.

## 1.1 Our Contribution

In the single-item setting, our research highlights the uniqueness of the Score-Only Mechanism as the sole deterministic, incentive-compatible, and monotone mechanism under mild conditions. Despite its simplicity, we discover that the additive reward gap between the Score-Only Mechanism and the optimal reward remains bounded. To enrich the choices available to the owner, we introduce a set of mechanisms offering two options. Notably, the optimal Two Menu mechanism can be efficiently computed in polynomial time. Additionally, we compute the optimal mechanism with a bounded menu size, shedding light on the tradeoff between menu size and reward. Surprisingly, we find that the collector's expected reward can increase without bounds as the menu size grows, and having a small menu size cannot guarantee any positive fraction of the optimal reward.

In the multi-item setting, we unveil the limitations of an ordinal mechanism that relies solely on the owner's item ordering as input, as it proves to be non-incentive-compatible. To address this, we propose a set of Union mechanisms that ingeniously combine single-item mechanisms.

Furthermore, we conduct experiments to assess the robustness of these mechanisms concerning the accuracy of the independent appraiser's assessments and the rate of items being acquired. These experiments shed valuable insights into the practicality and effectiveness of the proposed mechanisms in real-world scenarios.

## 1.2 Related work

Handling the fast-growing number of submissions to academic conferences has attracted a lot of attention. Almost all previous peer review mechanisms only rely on the reviewers' expertise [8, 17, 27]. To the best of our knowledge, the first exception was [35, 36], which recommends using the author's knowledge to assist in peer-reviewing because an author knows her papers the best. Su proposes the Isotonic Mechanism, which asks the author to report the ranking of her submissions. Wu et al. [39] subsequently extends the Isotonic Mechanism from the single-owner scenario to accommodate multiple owners, a scenario common in peer-review situations involving papers with multiple authors. The mechanism

returns adjusted scores by solving a convex optimization problem related to the ranking and imprecise raw scores given by reviewers. If the author's goal is to improve the accuracy of the scores for each paper, the author should provide accurate ranking information. While we will also consider asking the author (the owner of research papers) to provide a ranking of her papers, we consider the setting in that the owner wants to maximize the probability of the items getting acquired by a collector.

Miller et al. at 2005 introduce the peer-prediction problem and proposes a peer-prediction method to elicit informative feedback in online review platforms that can improve the accuracy of ratings and reduce the problem of shilling [28]. Faltigs and Radaovic at 2017 notice that in many data science applications, such as crowdsourcing, peer review, or online auctions, there is a problem of information asymmetry, where individuals have different levels of information about a particular item or decision. This can lead to biased or inaccurate information and undermine the data's usefulness [9]. They employ auction design to elicit truthful information from individuals. Dasgupta and Ghosh at 2013 propose a new approach to crowdsourced judgment elicitation that considers the crowd workers' varying levels of expertise [7]. The peer-prediction problem has also been studied in [13, 23–25, 33]. The major difference between our model and theirs is that we assume that the independent appraiser is non-strategic. The appraiser performs her duties with impartiality, which genuinely provides a score for the collector's item-acquiring decision, although the score may be subject to random noises.

Brier et al. at 1950 introduces the Brier score as a way to evaluate probabilistic forecasts [5]. The Brier score is a proper scoring rule that measures the accuracy of a probabilistic forecast by comparing the predicted probabilities to the actual outcomes. Examples of the scoring rule include logarithmic, quadratic, and spherical scoring rules [16, 18, 19, 31]. They are initially used in weather forecasting [6, 30], and have now been extended to prediction markets [14], finance, and macroeconomics [15, 29]. In our model, the goal is to optimize the collector's reward instead of just eliciting the true information.

The menu size is an important consideration in mechanism design as it can impact the optimality and practicability of the mechanism. A larger menu can provide more options for the participants, but it can also increase the complexity of the mechanism and reduce practicality. Therefore, researchers have proposed mechanisms of different menu sizes and examined their optimality in various settings [4, 21, 38].

## 2 SINGLE-ITEM MECHANISMS

An owner has an item whose factual quality, $v \in V$, is the owner's private information. The factual quality $v$ follows a discrete probability distribution $D$. Let $d(v)$ be the probability that the factual quality is $v$ and the support of $V$ be $|V| = n$. In a single-item acquiring mechanism, the owner declares a quality $v' \in V$ of her item, which is not necessarily the same as $v$. The collector will arrange for the item to be assessed by an independent appraiser and deemed a quality score. This quality score is subject to random noises and may not accurately reflect the item's factual value. Denote $r(v, s)$ the probability that the item's factual quality is $v$, and

it receives score $s$, where $s \in S$ and the support of $S$ is $|S| = m$. Let $R = [r(v, s)]$ be the stochastic matrix whose elements are $r(v, s)$ and $\sum_{s \in S} r(v, s) = 1, \forall v \in V$. The matrix $R$ and the distribution $D$ are public information. Denote $t$ the quality bar of this *item-acquiring problem*, which is chosen by nature. The quality bar influences the collector's decision-making since the collector has a positive reward if she acquires an item whose factual value $v$ exceeds $t$. Given the distribution $D$, the stochastic matrix $R$, and the quality bar $t$, the collector publicizes an acquiring matrix $X = [x(v', s)]$, in which the element $x(v', s)$ is the probability she acquires the item when the owner reports quality $v'$ and receives score $s$. The owner, in view of the acquiring matrix $X$ and the quality bar $t$, is interested in maximizing the probability that her item gets acquired by the collector. Therefore, she will report a quality $v'$ that maximizes

$$\sum_{s \in S} x(v', s) \, r(v, s).$$

When $x(v', s) \in \{0, 1\}$, we call the single-item acquiring mechanism *deterministic*. When $x(v', s) \in [0, 1]$, the mechanism is *randomized*. We are interested in *incentive-compatible* mechanisms that elicit truthful information from the owner. Incentive-compatibility is a crucial property in mechanism design because it ensures that individuals have no incentive to misrepresent their private information. Formally speaking, a mechanism is incentive-compatible if the owner maximizes the probability that her item gets acquired by declaring the true quality $v$, i.e.,

$$\sum_{s \in S} x(v, s)r(v, s) \geq \sum_{s \in S} x(v', s)r(v, s), \quad \forall v, v' \in V.$$

When the owner reports truthfully, the collector's reward is

$$\sum_{s \in S, v \in V} (v - t) \, d(v) \, x(v, s) \, r(v, s).$$

So, the collector's objective is to determine an acquiring matrix $X$ that elicits true information from the owner to maximize her reward. We call the mechanism that maximizes the collector's expected reward *optimal*.

As a principle, the collector relies on the expertise of the appraiser, so it is desirable that the acquiring matrix $X = [x(v, s)]$ is *monotone* increasing in the quality score $s$ for any fixed factual quality $v$. In other words, a mechanism is monotone if the higher quality of the item is considered by the independent appraiser, the more likely that the collector will acquire the item.

We begin by examining a simple mechanism that makes the acquiring decision regardless of the owner's report.

**The Score-Only Mechanism (SOM).** For any score $s$, the collector acquires the item if the expected quality of the item conditioning on the score reaches the quality bar, and denies it otherwise, i.e.,

$$x(v, s) = \begin{cases} 1, & E_{v \in V}[v \mid s] \geq t, \quad \forall s \in S, \\ 0, & \text{otherwise.} \end{cases}$$

Since

$$E_{v \in V}[v \mid s] = \sum_{v \in V} v \cdot \Pr[v \mid s] = \frac{\sum_{v \in V} vd(v)r(v, s)}{\sum_{v \in V} d(v)r(v, s)},$$

the acquiring matrix of the Score-Only Mechanism is such that $x(v, s) = 1$ if and only if

$$\sum_{v \in V} (v - t)d(v)r(v, s) \geq 0, \quad \forall s \in S.$$

In view that SOM makes the acquiring decision independent of the owner's report $v'$, the mechanism is incentive-compatible. In addition, the extent to which SOM maximizes the collector's reward depends on the stochastic matrix $R$. We call matrix $R$ *consistent* with probability distribution $D$, if for any $s$,

$$E_{v \in V}[v \mid s] \geq t \Leftrightarrow s \geq t.$$

Hence, with the consistency condition, the collector in SOM can acquire the item if the quality score deemed by the appraiser exceeds the quality bar, i.e., $s \geq t$. In practice, this criterion significantly facilitates the collector's decision-making. In the following, we show that SOM is the optimal mechanism that fulfills these desired properties. Due to space limitations, we defer some proofs in the paper to the Appendix.

THEOREM 2.1. *When the stochastic matrix $R$ is consistent with the quality distribution $D$, SOM is optimal amongst all deterministic, incentive-compatible, and monotone mechanisms.*

PROOF. Due to the consistency condition and that SOM is deterministic, the collector acquires the item if and only if $s \geq t$. Therefore, SOM is monotone in the quality score $s$. Next, we prove its optimality. Since the elements of the acquiring matrix, $X$, are binary in deterministic mechanisms, when we further restrict to monotone mechanisms, we know that for each $v_i \in V$, there exists an integer $k_i$, such that $x(v_i, s_1) = \cdots = x(v_i, s_{k_i}) = 0$, and $x(v_i, s_{k_i+1}) = \cdots = x(v_i, s_m) = 1$. We claim that due to incentive compatibility, this integer $k_i$ should be independent of the row $i$. Suppose for contradiction that there exist two rows $i$ and $j$, which correspond to two different factual qualities $v_i$ and $v_j$, for which $k_i \neq k_j$. Without loss of generality, assume that $k_i < k_j$. Then, when the item's factual quality is $v_j$, the owner can increase the probability that the item gets acquired by misreporting $v_i$, since $x(v_i, s_{k_i+1}) = 1$ whereas $x(v_j, s_{k_i+1}) = 0$, which violates incentive compatibility. Hence, the row rank of the acquiring matrix $X$ is 1. Last, the consistency condition uniquely determines the number of 1's in the rows of the acquiring matrix, except the ties when $s = t$, which does not make a difference to the collector's expected reward, no matter whether the mechanism acquires or denies the item. Therefore, SOM is optimal. □

While these properties are desirable, they restrict the search space to improve the collector's expected reward. We call a mechanism *omniscient* if it knows the factual quality $v$ and only acquires the item if $v \geq t$ and denies it otherwise. So, the omniscient mechanism is not confronted with these desirable properties. We first characterize the reward gap between SOM and the omniscient mechanism.

Define the sum of the differences between the factual quality and the score by *total bias*. That is, the total bias is

$$\sum_{s \in S, v \in V} |s - v| \, d(v)r(v, s).$$

We establish the following result.

THEOREM 2.2. *The difference between the collector's reward in the Omniscient Mechanism and her expected reward in the Score-Only Mechanism is bounded by the total bias.*

PROOF. Given a fixed $s \in S$, the quality deemed by the appraiser, we consider the difference between the collector's reward in the Omniscient Mechanism and her reward in the Score Only Mechanism. Their difference is due to that the SOM may not acquire the item while the Omniscient Mechanism does, when the factual quality is higher than the quality bar. We define it by the *loss conditioning on* $s$. That is,

$$L_s = \sum_{v \geq t} (v - t) d(v) r(v, s).$$

If SOM does not acquire the item, it is because that

$$\sum_{v \in V} (v - t) d(v) r(v, s) \leq 0. \tag{1}$$

Consider mutually exclusive cases $v < t$ and $v \geq t$, and rearrange this inequality, we have that

$$L_s \leq - \sum_{v < t} (v - t) d(v) r(v, s).$$

When $s \leq t$, we have

$$L_s \leq \sum_{v \geq t} (v - s) d(v) r(v, s) \leq \sum_{v \in V} |v - s| d(v) r(v, s). \tag{2}$$

When $s > t$, from (1), we obtain

$$\begin{aligned}
L_s &\leq - \sum_{v < t} (v - t) d(v) r(v, s) \\
&= - \sum_{v < s} (v - s) d(v) r(v, s) + \sum_{t \leq v < s} (v - s) d(v) r(v, s) \\
&\quad - \sum_{v < t} (s - t) d(v) r(v, s) \\
&\leq \sum_{v \in V} |v - s| d(v) r(v, s).
\end{aligned} \tag{3}$$

Combining function (2) with (3), we get

$$L_s \leq \sum_{v \in V} |s - v| d(v) r(v, s).$$

This is the total scoring bias conditioned on $s$. Summing over all $s$, we get that

$$L = \sum_{s \in S} L_s \leq \sum_{s \in S, v \in V} |s - v| d(v) r(v, s).$$

The proof for the mechanism acquires the item is relatively similar. Therefore, the difference between the two mechanisms is bounded by the total bias. □

Clearly, in order to improve the collector's expected reward, we need to relax these properties. In the following, we present a randomized yet simple mechanism. We start by proposing a set of these mechanisms and then find the optimal one within this set.

Let $\alpha \in [0, 1]$ and $b_1, b_2 \in S$, where $b_1 \leq b_2$. Given any fixed factual quality $v \in V$, due to monotonicity, assume that the scores $s_j \in S$, $j \in [m]$ are in increasing order. Let us consider two types of acquiring vectors as below.

$$x_1(v, s) = \begin{cases} 0, & \text{for } s_1 < \cdots < s_l < b_1, \\ \alpha, & \text{for } b_1 \leq s_{l+1} < \cdots < s_m, \end{cases}$$

$$x_2(v, s) = \begin{cases} 0, & \text{for } s_1 < \cdots < s_h < b_2, \\ 1, & \text{for } b_2 \leq s_{h+1} < \cdots < s_m. \end{cases}$$

Recall that the owner wants to maximize the probability that the item gets acquired by the collector; that is, to maximize the $\sum_{s \in S} x(v', s) r(v, s)$. Since $R = r(v, s)$ is public information, one can solve this maximization problem on behalf of the owner when rows of the acquiring matrix are restricted to the above two types. Essentially, this problem boils down to comparing $\alpha \sum_{s \geq b_1} r(v, s)$ and $1 \cdot \sum_{s \geq b_2} r(v, s)$. If $\alpha \sum_{s \geq b_1} r(v, s) > 1 \cdot \sum_{s \geq b_2} r(v, s)$, then the acquiring vector $x_1(v, s)$ is preferable; otherwise, $x_2(v, s)$ is preferable. Denote $V_1 = \{v \mid \alpha \sum_{s \geq b_1} r(v, s) > \sum_{s \geq b_2} r(v, s)\} \subset V$ the set of the owner's factual qualities with which $x_1(v, s)$ is preferable, and $V \backslash V_1$ the set of factual qualities with which $x_2(v, s)$ is preferable.

**Two Menu Mechanisms (TMM).** A Two Menu Mechanism (TMM($b_1, b_2, \alpha$)) offers two options, namely menus, to the owner. In other words, the row rank of the acquiring matrix is 2. Specifically, the acquiring matrix is

$$x(v, s) = \begin{cases} 0, & \text{if } v \in V_1 \text{ and } s < b_1, \\ \alpha, & \text{if } v \in V_1 \text{ and } s \geq b_1, \\ 0, & \text{if } v \notin V_1 \text{ and } s < b_2, \\ 1, & \text{if } v \notin V_1 \text{ and } s \geq b_2. \end{cases}$$

So, one menu is that the collector will acquire the item with probability $\alpha$ if the quality score $s$ reaches $b_1$, and denies it otherwise. The other menu is that the collector will acquire the item if the quality score $s$ reaches $b_2$, and denies it otherwise.

Intuitively, the owner prefers the first menu if the item is deemed low quality by the appraiser and prefers the second menu if it is deemed high quality.

THEOREM 2.3. *Any Two Menu Mechanism is incentive-compatible and monotone. The optimal Two Menu Mechanism can be found in $O(m^3 n \log n)$ time.*

PROOF. A Two Menu Mechanism is incentive compatible because the owner's most favorable action is captured by the definition of sets $V_1$ and $V \backslash V_1$. The monotonicity can be seen from the acquiring probabilities $x(v, s)$.

Consider two parameters $b_1, b_2 \in S$ in a Two Menu Mechanism. Since $|S| = m$, the total number of these pairs of parameters is $m^2$. For each pair of $b_1$ and $b_2$, we can find the probability $\alpha$ that maximizes the collector's expected reward. These three parameters $b_1, b_2, \alpha$ define a Two Menu Mechanism. We find the optimal Two Menu Mechanism by comparing the collector's expected reward in all $m^2$ of these Two Menu Mechanisms.

Without loss of generality, assume that the quality values in $V$ are sorted in increasing order, i.e., $v_1 < v_2 < \cdots < v_n$. For each pair of $b_1$ and $b_2$, when $\alpha = 0$, $V_1 = \emptyset$. The cardinality of the set $V_1$ increases when $\alpha$ increases. For every $i = 1, \cdots, n$, we compute the probability $\alpha_i$ such that the quality value $v_i$ is added to

$V_1$. We sort these $\alpha_i$'s in non-decreasing order and denote them by $\alpha_i^{(1)}, \cdots, \alpha_i^{(n)}$. It takes time $O(n \log n)$. For any $\alpha \in [\alpha_i^{(j)}, \alpha_i^{(j+1)})$, $j = 1, \cdots, n-1$, $V_1$ is determined. The collector's expected reward becomes a linear function of $\alpha$, and it takes $O(1)$ time to find the optimal $\alpha$ in the interval $[\alpha_i^{(j)}, \alpha_i^{(j+1)})$. After searching in all intervals, we pick the best $\alpha$. To conclude, the time complexity is $O(m^3 n \log n)$. □

If we further relax the menu size, we can derive the optimal mechanisms by solving a linear programming.

**The Optimal Mechanisms (OM$_1$).** The acquiring matrix $X = x(v, s)$ of the optimal mechanisms that maximize the collector's expected reward is the solution to the following linear programming.

$$\max \sum_{s \in S, v \in V} (v - t) \, d(v) \, x(v, s) \, r(v, s)$$

$$s.t. \sum_{s \in S} x(v, s) r(v, s) \geq \sum_{s \in S} x(v', s) r(v, s), \quad \forall v, v' \in V, \quad (4)$$

$$x(v, s) \geq x(v, s'), \quad \forall v \in V, s > s' \in S,$$

$$x(v, s) \in [0, 1], \quad \forall v \in V, s \in S.$$

We note that the optimal solution to this linear programming may not be unique, and any optimal solution $X = [x(v, s)]$ defines an Optimal Mechanism.

THEOREM 2.4. *Any Optimal Mechanism derived from the optimal solution of the linear programming is incentive compatible and monotone. The set of optimal mechanisms is convex.*

PROOF. The first constraint of the linear programming implies that the owner has a better chance of getting the item acquired by reporting the factual quality $v$ than by misreporting any other value $v'$. Therefore, it is incentive compatible. The second constraint implies that the acquiring vectors $x(v, s)$ are monotone in score $s$.

We notice that any linear combination of a set of feasible solutions to the linear programming is feasible. Hence, any linear combination of a set of optimal solutions is feasible and optimal. So, the set of optimal mechanisms is convex. □

We provide an $OM_1$ as an example.

*Example 2.5.* The set of possible factual quality is $V = S = \{0, \frac{1}{3}, \frac{2}{3}, 1\}$ and the probabilities of the values are $d(0) = 0.264$, $d(\frac{1}{3}) = 0.539$, $d(\frac{2}{3}) = 0.186$, $d(1) = 0.012$. The quality bar is $t = 0.5$. The stochastic matrix $R$ and the acquiring matrix $X$ by solving the linear programming shown below:

$$R = \begin{bmatrix} 0.762 & 0.122 & 0.072 & 0.044 \\ 0.009 & 0.792 & 0.136 & 0.063 \\ 0.038 & 0.127 & 0.825 & 0.010 \\ 0.031 & 0.052 & 0.171 & 0.746 \end{bmatrix},$$

$$X = \begin{bmatrix} 0.044 & 0.044 & 0.044 & 0.044 \\ 0.0 & 0.0 & 0.37931 & 0.37931 \\ 0.0 & 0.0 & 0.37931 & 0.37931 \\ 0.0 & 0.0 & 0.0 & 1.0 \end{bmatrix}.$$

In this example, the row rank of the acquiring matrix $X$ is three, and each linearly independent row can be viewed as a different menu to the owner [21, 38]. Since simple mechanisms, in terms of

providing fewer menus for the owner to choose from, are preferable, we bound the size of menus in an optimal mechanism as below.

THEOREM 2.6. *Denote $V_B$, the set of factual qualities with a value above the quality bar $t$. That is, $V_B = \{v \mid v \in V, v > t\}$. There exists an optimal mechanism $OM_1$ whose menu size is at most $|V_B|$.*

PROOF. Suppose $X = [x(v, s)]$ is an optimal solution to the optimization problem. For any $v \leq t$, we define a mapping $f$ as

$$f(v) = \arg \max_{\bar{v} > t} \sum_{s \in S} x(\bar{v}, s) r(v, s).$$

Given any $v \leq t$, which corresponds to one menu $x(v, s)$, $f(v)$ maps it to a value above the quality bar, which corresponds to an existing menu of the optimal mechanism defined by $X = [x(v, s)]$. Therefore, we have a new mechanism $\hat{X} = \hat{x}(v, s)$, where

$$\hat{x}(v, s) = \begin{cases} x(f(v), s), & \text{if } v \leq t, \\ x(v, s), & \text{if } v > t. \end{cases}$$

Clearly, the menu size of the new mechanism $\hat{X}$ is bound by $|V_B|$. In addition, $\hat{x}(v, s)$ satisfies the linear programming constraints, so it is indeed a feasible solution. For optimality, since the mapping $f(v)$ reduces the feasible region when $v \leq t$, the acquiring probability becomes weakly smaller, i.e., $\sum_{s \in S} x(v, s) r(v, s) \geq \sum_{s \in S} \hat{x}(v, s) r(v, s), \forall v \leq t$. Therefore, the objective value of the linear programming becomes weakly larger. For $v > t$, nothing changes. In all, $\hat{X} = [\hat{x}(v, s)]$ is an optimal mechanism whose menu size is at most $|V_B|$. □

Finally, we compare the expected reward of the collector in the above mechanisms. For this purpose, the approximation ratio, which originated from theoretical computer science, turns out to be a compelling language [3, 20, 22]. In our context, the *approximation ratio* is the largest ratio between the expected reward from two mechanisms.

THEOREM 2.7. *The approximation ratio of TMM to SOM, is unbounded. The approximation ratio of $OM_1$ to TMM, is unbounded.*

In other words, considering the menu size of SOM, TMM, and $OM_1$ are 0, 2, and at most $|V_B|$, respectively, these ratios imply that simple mechanisms with a small menu size cannot ensure any positive fraction of the optimal reward of mechanisms with a larger menu size.

The proof is by constructing instances that an unbounded ratio between two mechanisms is attained. The proof is deferred to the Appendix.

## 3 MULTI-ITEM MECHANISMS

In the multi-item setting, the owner has $k$ items. Denote the factual quality of item $i$ by $v_i \in V$, $i \in [k]$, and the quality vector $\vec{v} = (v_1, \cdots, v_k)$. Each factual quality $v_i$ independently and identically follows the discrete probability distribution $D$. Denote $r(v_i, s_i)$ the probability that item $i$'s factual quality is $v_i$, and it is deemed as score $s_i$ by the independent appraiser, where $s_i \in S, i \in [k]$, and the score vector $\vec{s} = (s_1, \cdots, s_k)$. The quality bar $t$ is item independent, i.e., the value of $t$ is the same for all items. The collector's acquiring decision is represented by $X = (X_1, \cdots, X_k)$, where $X_i = [x_i(\vec{v}, \vec{s})]$ is the acquiring matrix for item $i$. Therefore, the owner's objective

is to maximize the total probability of her items being acquired, which is

$$\sum_{\vec{s}} \sum_{i \in [k]} \left( x_i(\vec{v}, \vec{s}) \prod_{j \in [k]} r(v_j, s_j) \right).$$

**The Optimal Mechanisms (OM$_k$).** The acquiring matrices $X = (X_1, \cdots, X_k)$ of the optimal mechanisms that maximize the collector's expected reward is the solution to the following linear programming.

$$\max \sum_{\vec{v}, \vec{s}} \sum_{i \in [k]} \left( (v_i - t) x_i(\vec{v}, \vec{s}) \prod_{j \in [k]} r(v_j, s_j) d(v_j) \right)$$

$$s.t. \sum_{\vec{s}} \sum_{i \in [k]} \left( x_i(\vec{v}, \vec{s}) \prod_{j \in [k]} r(v_j, s_j) \right)$$

$$\geq \sum_{\vec{s}} \sum_{i \in [k]} \left( x_i(\vec{v}', \vec{s}) \prod_{j \in [k]} r(v_j, s_j) \right), \quad \forall i, \vec{v}', \vec{v}, \vec{s}, \quad (5)$$

$$x_i(\vec{v}, (\vec{s}_{-i}, s_i)) \geq x_i(\vec{v}, (\vec{s}_{-i}, s_i')), \quad \forall i, \vec{v}, \vec{s}_{-i}, s_i \geq s_i',$$

$$x_i(\vec{v}, \vec{s}) \in [0, 1], \qquad \forall i, \vec{v}, \vec{s}.$$

Similar to (4), this linear programming aims to maximize the collector's expected reward. The first constraint guarantees incentive compatibility of the mechanism, and the second constraint implies monotonicity. Although OM$_k$ retains these desired properties, the size of the linear programming grows exponentially in the number of items. As a matter of course, we strive to design simple mechanisms for ease of implementation. We start by exploring ordinal mechanisms that only take the owner's ranking of the items as input.

Su [35] considered an author-assisted peer-reviewing problem. They proposed a truthful mechanism that takes only the author's ranking of the papers to produce a more accurate cardinal grading. In light of this, we design a mechanism for our problem that takes only the owner's ranking of the items to maximize the collector's expected reward. For simplicity, we present the mechanism in the form of only two items and investigate the incentive compatibility of the mechanism.

**The Ranking Mechanism (RM).** Let the factual quality of the owner's two items be $v_1$ and $v_2$, and the owner's reports be $v_1'$ and $v_2'$. Denote $V_g = \{(v_1', v_2') | v_1' > v_2'\}$, $V_e = \{(v_1', v_2') | v_1' = v_2'\}$, $V_s = \{(v_1', v_2') | v_1' < v_2'\}$, and $V_{rank} = \{V_g, V_e, V_s\}$. Essentially, the mechanism only needs to take one of the elements in $V_{rank}$ as input, rather than the exact values of $v_1'$ and $v_2'$. Given the owner's reports, the Ranking Mechanism computes the conditional expected values of the factual quality, i.e., for $i = 1, 2$,

$$E\left[v_i | s_1, s_2, V_{rank}\right] = \frac{\sum_{(v_1, v_2) \in V_{rank}} v_i d(v_1) d(v_2) r(v_1, s_1) r(v_2, s_2)}{\sum_{(v_1, v_2) \in V_{rank}} d(v_1) d(v_2) r(v_1, s_1) r(v_2, s_2)}.$$

Then, the mechanism acquires item $i$ if the expected factual value conditioning on the ranking and deemed quality is greater than the quality bar. That is,

$$x_{i,rank}(s_1, s_2) = \begin{cases} 1, & E\left[v_i | s_1, s_2, V_{rank}\right] \geq t, \\ 0, & \text{otherwise.} \end{cases}$$

Finally, the total probability of acquiring both items when the owner reports $V_{rank}$ is

$$x_{rank} = \sum_{s_1, s_2 \in S} r(v_1, s_1) r(v_2, s_2)(x_{1,rank}(s_1, s_2) + x_{2,rank}(s_1, s_2)).$$

THEOREM 3.1. *The Ranking Mechanism RM is not incentive compatible.*

Next, we propose a class of mechanisms – the Union Mechanisms – each consists of single-item mechanisms (SOM, TMM, OM$_1$, and a combination of them or any other mechanisms). Recall $|V| = n$, and denote the $n$ discrete values in $V$ by $v^{(1)}, \cdots, v^{(n)}$.

**The Union Mechanisms (UM).** A Union Mechanism consists of $k$ independent single-item mechanisms, each applied to an item $i \in [k]$. Denote $Y_i = [y_i(v_i, s_i)]$ the acquiring matrix of item $i$. Let $\Gamma_y(\vec{v}, \vec{s}) = \sum_i y_i(v_i, s_i)$. There must exist a number $q \in [n]$ such that

$$\left| \{i \mid v_i > v^{(q)}\} \right| < \Gamma_y(\vec{v}, \vec{s}) \leq \left| \{i \mid v_i \geq v^{(q)}\} \right|. \quad (6)$$

The acquiring probability of item $i$ in the Union Mechanism is

$$x_i(\vec{v}, \vec{s}) = \begin{cases} 1, & \text{if } v_i > v^{(q)}, \\ \frac{\Gamma_y(\vec{v}, \vec{s}) - \left|\{i \mid v_i > v^{(q)}\}\right|}{\left|\{i \mid v_i = v^{(q)}\}\right|}, & \text{if } v_i = v^{(q)}, \\ 0, & \text{if } v_i < v^{(q)}. \end{cases}$$

We establish the following properties of any Union Mechanism.

THEOREM 3.2. *If each of the $k$ single-item mechanisms is incentive compatible, then the Union Mechanism is incentive compatible. If $y_i(v_i, s_i)$ is monotone in $s_i$, then $x_i(\vec{v}, \vec{s})$ is monotone in $s_i$, for any $\vec{v}$ and $\vec{s}_{-i}$.*

The Union Mechanism weakly improves the conference profit on every valuation and score profile. The idea is to reduce the likelihood of getting low-quality items and increase the likelihood of getting high-quality items. It always performs weakly better than we use mechanisms for single paper submissions independently.

Recall that SOM attains the smallest reward amongst several single-item mechanisms, and its reward gap with the omniscient mechanism is lower bounded. Therefore, in a multi-item setting, we can lower bound the reward gap between a Union Mechanism consisting of any single-item mechanisms with the omniscient mechanism. For example, we present the following two results regarding the collector's expected reward maximization.

COROLLARY 3.3. *The reward gap between the omniscient mechanism and the Union Mechanism consisting of $k$ Two Menu Mechanisms is lower bounded by $k$ times of total bias.*

PROOF. Essentially, a Union Mechanism aggregates the acquiring probabilities of $k$ single-item mechanisms and redistributes the probabilities amongst the $k$ items whose qualities are above the threshold value $v^{(q)}$ in an even manner. So, the composite Union Mechanism is incentive compatible when the underlying $k$ single-item mechanisms are incentive compatible.

For any $\vec{v}$ and $\vec{s}_{-i}$, when $s_i$ increases, then $y_i(v_i, s_i)$ increases, hence $\Gamma_y(\vec{v}, \vec{s})$ increases. So, $x_i(\vec{v}, (\vec{s}_{-i}, s_i))$ increases, since it is either independent of $s_i$ or increasing in $s_i$. □

THEOREM 3.4. *In the $k$-item setting, the approximation ratio of the optimal multi-item mechanism $OM_k$ to the Union Mechanism consisting of $k$ optimal single-item mechanisms $OM_1$ is unbounded. The approximation ratio of the Union Mechanism consisting of $k$ optimal single-item mechanisms $OM_1$ to the sum of the collector's rewards in $k$ optimal single-item mechanisms $OM_1$ is unbounded.*

## 4 EXPERIMENTAL RESULTS

In the preceding sections, we devoted our efforts to designing mechanisms for the item-acquiring problem, with a focus on optimizing the collector's expected reward while ensuring specific desirable properties. In this section, we present the results of our experiments, aimed at achieving two key objectives. Firstly, we assess the robustness of these mechanisms concerning the independent appraiser's assessment accuracy. Secondly, we investigate the impact of these mechanisms on the items' acquiring rate when put into practice. The item-acquiring problem bears striking similarities to the peer-reviewing process prevalent in the academic world [35, 36]. To gain insights into the effectiveness of these mechanisms in a real-world academic context, we conduct experiments to evaluate how well they perform when reviewers' assessments are subject to random noises. Additionally, we analyze the implications of deploying different mechanisms on the item acquiring rate, shedding light on their practicality and efficacy in this academic setting. These experiments allow us to better understand the behavior of the mechanisms under various scenarios, providing valuable insights for potential improvements and real-world applications.

### 4.1 Experimental Setting

We set $V = S = \{0, \frac{1}{6}, \frac{2}{6}, \frac{3}{6}, \frac{4}{6}, \frac{5}{6}, 1\}$ and $t = 0.25$. We consider two scenarios: one involving the acquisition of a single item and the other involving the acquisition of two items, i.e., $k = 2$. For our experiments, we vary two parameters: the distribution $D$ that the factual quality $v$ follows and the distribution $r(v, \cdot)$ when $v$ is fixed. We compare these mechanisms when these probabilities follow normal and log-normal distributions.

**Normal distribution.** We set $D \sim N(0.3, 0.25)$. We discretize the normal distribution by dividing the real axis into seven intervals and scale the probabilities of values falling into each interval so that they sum to 1. That is, $d(0) = 0.1377, d(\frac{1}{6}) = 0.245, d(\frac{2}{6}) = 0.2804, d(\frac{3}{6}) = 0.2054, d(\frac{4}{6}) = 0.0968, d(\frac{5}{6}) = 0.0291, d(1) = 0.0057$. For each $v \in V$, we let $r(v, \cdot)$ follow normal distributions with the mean value $v$; the variance varies within the range $[0, 0.6]$ and has a step size of 0.001.

**Log-normal distribution.** In practice, the log-normal distribution is commonly used to model systems' reliability [26]. In particular, it models the failure rates of complex systems that have multiple failure modes, each with its own distribution. This is particularly relevant to our problem since the quality of an item can be influenced by a wide range of factors, such as material quality and manufacturing processes. For a direct comparison with the normal distribution, we choose the same mean and variance values.

### 4.2 Results

For single-item acquiring, we can compute the acquiring matrix of the Score-Only Mechanism (SOM) when the distributions are

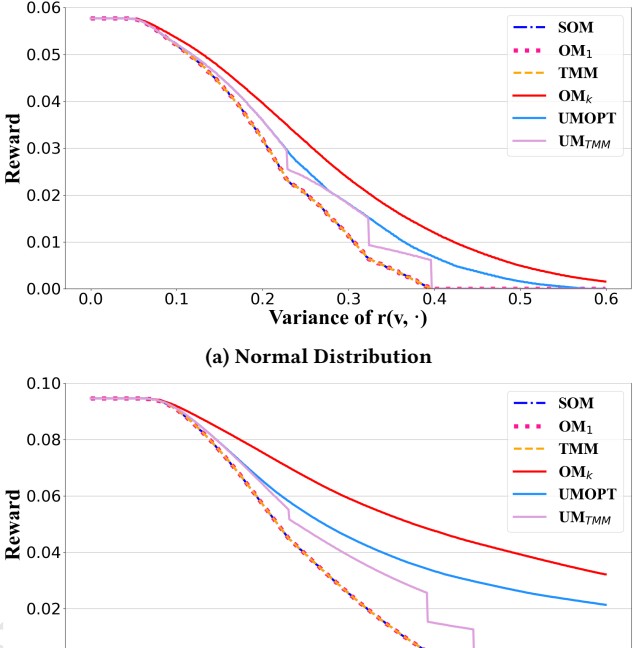

**(a) Normal Distribution**

**(b) Log-Normal Distribution**

Figure 1: Subfigures (a) and (b) illustrate the collector's expected reward when $D$ and $r(v, \cdot)$ follow the normal and log-normal distributions, respectively. For multiple-item mechanisms, the figures show the collector's expected reward divided by the number of items.

fixed. For Two Menu Mechanisms (TMM), we compute the optimal $\alpha, b_1, b_2$ that maximizes the collector's expected reward and compare this optimal TMM with other mechanisms. In the case of Optimal Mechanisms ($OM_k$), there may be multiple optimal solutions and we select the solution provided by Gurobi, a linear programming solver. For multiple-item acquiring, we consider two types of Union Mechanisms. One is $UM_{TMM}$, which is the Union Mechanism consisting of $k$ Two Menu Mechanisms. The other is UMOPT, which is the optimal Union Mechanism that consists of $k$ single-item mechanisms.

**Collector's expected reward.** We first look at how these mechanisms perform when $D$ and $r(v, \cdot)$ follow normal distributions (Figure 1a). For the single-item case, we notice that the collector's expected reward drops when the variance of the normal distribution $r(v, \cdot)$ increases, and the difference between three single-item mechanisms, SOM, $OM_1$, and TMM, is very small. In the multiple-item case, we compute the collector's expected reward of the multiple items and show the average reward (per item) in the same figure. We observe that the greater the accuracy of the independent appraiser's assessment, the higher the collector's anticipated reward. Furthermore, the collector's expected reward is higher when employing a multi-item mechanism, such as $OM_k$, UMOPT, or $UM_{TMM}$, compared to using a single-item mechanism multiple times. When $D$

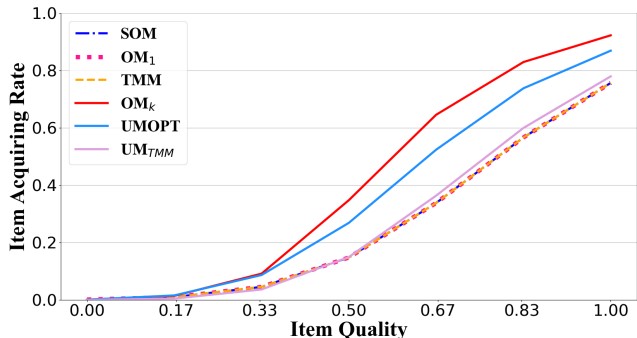

(a) Normal Distribution

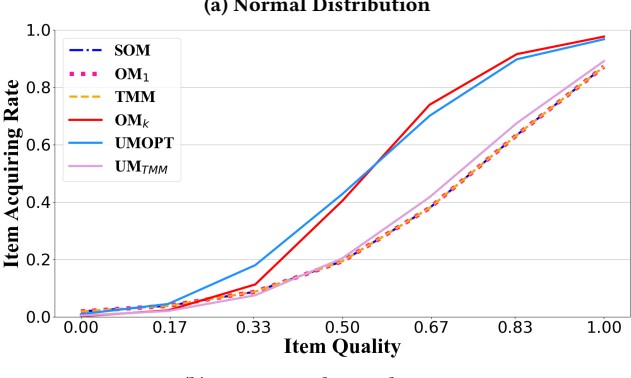

(b) Log-Normal Distribution

**Figure 2: Subfigures (a) and (b) illustrate the item acquiring rate when $D$ and $r(v, \cdot)$ follow the normal and log-normal distributions, respectively.**

and $r(v, \cdot)$ follow log-normal distributions (Figure 1b), we obtain similar results.

**Item acquiring rate.** Figure 2a presents our results in terms of the item acquiring rate when $r(v, \cdot)$ follows the normal distribution $N(v, 0.25)$ for any $v \in V$. It is important to note that the $\text{UM}_{TMM}$ outperforms the single-item mechanism regardless of the quality of the item. Compared to TMM, it reduces the likelihood of getting low-quality items and increase the likelihood of getting high-quality items. The results for log-normal distributions are presented in Figure 2b. The observed trends closely resemble those observed in the case of normal distributions.

## 5 CONCLUSION AND FUTURE WORK

In our research, we introduced a compelling item-acquiring problem in which the owner holds confidential information about item quality. Her goal is to maximize the collector's chances of acquiring these items, while the collector, armed with the appraiser's assessment and public data, aims to maximize the expected reward from acquisitions. The inherent information asymmetry between them posed a challenge: designing incentive-compatible mechanisms to encourage the owner's truthful disclosure of item quality. To address this, we proposed various deterministic and randomized mechanisms for single-item and multi-item scenarios, each with different menu sizes. By rigorously evaluating and comparing these mechanisms, we aimed to identify effective strategies for eliciting

truthful information from the owner and optimizing the acquisition process. Our study sheds light on the intricate dynamics between owners and collectors in the digital marketplace, offering insights for improving real-world item-acquiring scenarios.

We addressed this challenge as a mechanism design problem without payments. Its fundamental nature distinguishes itself from classic mechanism design problems. In future work, we can explore it as a mechanism design problem with payments, which could have potential applications in various domains, as indicated below.

*The used car markets.* Akerlof's theory of the "Market for Lemons" serves as a classic illustration of how information asymmetry can significantly impact market outcomes [2]. Specifically, in this scenario, the seller of a low-quality car (a "lemon") is aware of its true condition, while the seller of a high-quality car possesses knowledge about its superior quality. Since buyers lack the ability to easily distinguish between the two types of cars, an information gap arises between buyers and sellers. Consequently, buyers are more inclined to purchase low-quality cars because they cannot differentiate them from high-quality ones, leading to adverse selection. This, in turn, can lead to market failure. Akerlof's theory proposes that a potential solution to this problem is to devise mechanisms that encourage sellers to reveal the true quality of their products. By finding ways to elicit genuine information from sellers, we can mitigate the adverse effects of information asymmetry and improve market efficiency. This theory continues to be relevant and influential in the study of market dynamics and strategies to address information imbalances.

*Antique collection.* The quest for rare and unique items is a competitive and demanding endeavor. Antique collectors frequently turn to independent third-party experts for guidance when evaluating the value and quality of potential acquisitions. These experts can be appraisers, dealers, auction houses, or specialized consultants with in-depth knowledge in a specific category of antiques. An independent appraisal offers collectors an objective and impartial assessment of an item's authenticity and condition. Nonetheless, it's important to note that the owner of an antique item usually possesses more detailed knowledge about its provenance and unique characteristics compared to a collector and appraiser. This information asymmetry between the owner and the collector can present unique challenges when evaluating and acquiring these treasured artifacts.

In these applications, there is a monetary exchange between the seller and the buyer, which differs from the relationship between the item owner and the collector. In the case of the seller, the goal may be to maximize revenue, while the buyer may have quasi-linear utilities and budget constraints.

Another fascinating avenue for exploration is to approach this problem through the lens of the learning-augmented mechanism design framework. This approach leverages predictions to enhance decision-making, as demonstrated in previous works [1, 12]. Incorporating predictive elements could provide valuable insights and potentially optimize the mechanism further, warranting further investigation and potential breakthroughs in addressing this problem from different angles.

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

# APPENDIX: MISSING PROOFS

In this section, we provide the proofs omitted in the paper due to space limits.

## Proof of Theorem 2.7:

PROOF. First, we construct an instance to show that the approximation ratio of TMM to SOM, is unbounded. In this instance, we use the following settings to find special examples. $t = 0.5$, $V = \{0, 0.33, 0.66, 1\}$, $S = \{0, 0.33, 0.66, 1\}$. $d[0] = 0.262$, $d[\frac{1}{3}] = 0.535$, $d[\frac{2}{3}] = 0.191$, $d[1] = 0.012$. The matrix $R$ is constructed as follows:

$$R = \begin{bmatrix} 0.754 & 0.133 & 0.077 & 0.036 \\ 0.013 & 0.701 & 0.261 & 0.025 \\ 0.008 & 0.173 & 0.814 & 0.005 \\ 0.017 & 0.030 & 0.037 & 0.916 \end{bmatrix}.$$

In this instance, the acquiring matrix of the two mechanisms are that

$$X_{SOM} = \begin{bmatrix} 0.0 & 0.0 & 0.0 & 0.0 \\ 0.0 & 0.0 & 0.0 & 0.0 \\ 0.0 & 0.0 & 0.0 & 0.0 \\ 0.0 & 0.0 & 0.0 & 0.0 \end{bmatrix},$$

$$X_{TMM} = \begin{bmatrix} 0.0 & 0.0 & 0.0 & 1.0 \\ 0.0 & 0.0 & 0.08741 & 0.08741 \\ 0.0 & 0.0 & 0.08741 & 0.08741 \\ 0.0 & 0.0 & 0.0 & 1.0 \end{bmatrix}.$$

In this particular case, the collector's reward in $SOM$ is 0, while the collector's reward in $TMM$ is 0.0002075. Hence, the approximation ratio is unbounded.

Next, we construct an instance to show that the approximation ratio of $OM_1$ to TMM, is unbounded. In this instance, we use the following settings to find special examples. $t = 0.5$, $V = \{0, 0.33, 0.66, 1\}$, $S = \{0, 0.33, 0.66, 1\}$. $d[0] = 0.262$, $d[\frac{1}{3}] = 0.535$, $d[\frac{2}{3}] = 0.191$, $d[1] = 0.012$. The matrix $R$ is constructed as follows:

$$R = \begin{bmatrix} 0.71 & 0.13 & 0.11 & 0.05 \\ 0.03 & 0.82 & 0.09 & 0.06 \\ 0.11 & 0.13 & 0.72 & 0.04 \\ 0.01 & 0.08 & 0.15 & 0.76 \end{bmatrix}.$$

In this instance, the acquiring matrix of the two mechanisms are that

$$X_{OM_1} = \begin{bmatrix} 0.0 & 0.0 & 0.4 & 0.4 \\ 0.06 & 0.06 & 0.06 & 0.06 \\ 0.0 & 0.0 & 0.4 & 0.4 \\ 0.0 & 0.0 & 0.0 & 1.0 \end{bmatrix},$$

$$X_{TMM} = \begin{bmatrix} 0.0 & 0.0 & 0.0 & 0.0 \\ 0.0 & 0.0 & 0.0 & 0.0 \\ 0.0 & 0.0 & 0.0 & 0.0 \\ 0.0 & 0.0 & 0.0 & 0.0 \end{bmatrix}.$$

In this particular case, the collector's reward in $TMM$ is 0, while the collector's reward in $OM_1$ is 0.000503. Hence, the approximation ratio is unbounded. □

## Proof of Theorem 3.1

PROOF. By constructing a counterexample, we prove that the Ranking Mechanism is not incentive compatible. Our example shows that the owner can increase his expected utility by reporting an untruthful value ranking of two items.

The example is as follows:

Let $t = 0.5$, $V = \left\{0, \frac{1}{3}, \frac{2}{3}, 1\right\}$, $S = \left\{0, \frac{1}{3}, \frac{2}{3}, 1\right\}$, $d(0) = 0.262$, $d(\frac{1}{3}) = 0.535$, $d(\frac{2}{3}) = 0.191$, $d(1) = 0.012$,

$$R(v, s) = \begin{bmatrix} 0.84 & 0.12 & 0.02 & 0.02 \\ 0.14 & 0.80 & 0.05 & 0.01 \\ 0.07 & 0.18 & 0.72 & 0.03 \\ 0.06 & 0.08 & 0.14 & 0.72 \end{bmatrix}.$$

After computation, we can obtain:

$$x_{V_g} = \begin{bmatrix} 0.1724 & 0.8510 & 0.8320 & 0.2372 \\ 0.1758 & 0.8195 & 0.3372 & 0.1562 \\ 0.7820 & 0.9550 & 0.8670 & 0.7840 \\ 0.8924 & 1.0110 & 1.4468 & 0.9804 \end{bmatrix},$$

$$x_{V_e} = \begin{bmatrix} 0.0032 & 0.0048 & 0.0600 & 0.0688 \\ 0.0048 & 0.0072 & 0.0900 & 0.1032 \\ 0.0600 & 0.0900 & 1.1250 & 1.2900 \\ 0.0688 & 0.1032 & 1.2900 & 1.4792 \end{bmatrix},$$

$$x_{V_s} = \begin{bmatrix} 0.1724 & 0.1758 & 0.7820 & 0.8924 \\ 0.8510 & 0.8195 & 0.9550 & 1.0110 \\ 0.8320 & 0.3372 & 0.8670 & 1.4468 \\ 0.2372 & 0.1562 & 0.7840 & 0.9804 \end{bmatrix}.$$

A counter-case can be identified in the preceding examples to demonstrate that the mechanism lacks truthfulness. Specifically, when $v_1$ equals $\frac{2}{3}$ and $v_2$ equals 0, we observe that $x_{V_g}(\frac{2}{3}, 0) = 0.7820$, which is less than $x_{V_s}(\frac{2}{3}, 0) = 0.8320$. This example indicates that the probability of accepting the sum of two items that truthfully disclose $V_g$ is 0.782, but if the Owner provides false information by disclosing $V_s$, then the probability of accepting two items increases to 0.832. Thus, the mechanism is not incentive-compatible. □

## Proof of Theorem 3.4:

First, we construct an instance to show that the approximation ratio of $OM_k$ to $UM_{OM_1}$ is unbounded. To simplify the complex situation, we assume that the owner has only two items. Then we compare the collector's expected reward in these two mechanisms.

We set $t = 0.5$, $V = \{0, 0.33, 0.66, 1\}$, $S = \{0, 0.33, 0.66, 1\}$. $d[0] = 0.2645$, $d[\frac{1}{3}] = 0.5386$, $d[\frac{2}{3}] = 0.1861$, $d[1] = 0.0109$. The matrix $R$ is constructed as follows:

$$R = \begin{bmatrix} 0.522 & 0.232 & 0.145 & 0.101 \\ 0.022 & 0.708 & 0.221 & 0.049 \\ 0.004 & 0.427 & 0.515 & 0.054 \\ 0.066 & 0.113 & 0.270 & 0.551 \end{bmatrix}.$$

Therefore, the collector's expected reward are 0 for the objective of $UM_{OM_1}$ and 0.0085264 for $OM_2$. Thus, we can conclude that the approximation ratio is unbounded. For a detailed account of the results, please refer to the following anonymous URL due to page constraints: https://anonymous.4open.science/r/On-Truthful-Item-Acquiring-Mechanisms-for-Reward-Maximization-FD9C

Next, we construct an instance to show that the approximation ratio of $UM_{OM_1}$ to $k \times OM_1$ is unbounded. To simplify the complex

situation, we assume that the owner has only two items. Then we compare the collector's expected reward in these two mechanisms.

We set $t = 0.5$, $V = \{0, 0.33, 0.66, 1\}$, $S = \{0, 0.33, 0.66, 1\}$. $d[0] = 0.2645$, $d[\frac{1}{3}] = 0.5386$, $d[\frac{2}{3}] = 0.1861$, $d[1] = 0.0109$. The matrix $r$ is constructed as follows:

$$R = \begin{bmatrix} 0.749 & 0.128 & 0.074 & 0.049 \\ 0.057 & 0.737 & 0.190 & 0.016 \\ 0.018 & 0.086 & 0.834 & 0.062 \\ 0.144 & 0.147 & 0.209 & 0.500 \end{bmatrix}.$$

Therefore, the collector's expected reward is 0 for the objective of $2 \times OM_1$ and 0.0248746 for $UM_{OM_1}$. Thus, we can conclude that the approximation ratio is unbounded. For a detailed account of the results, please refer to the following anonymous URL, which has been provided due to page limitations: https://anonymous.4open.science/r/On-Truthful-Item-Acquiring-Mechanisms-for-Reward-Maximization-FD9C

**Code from experiment:**

The experimental section of this paper employs Python code, utilizing Gurobi as a solver for both linear programming and integer programming tasks. For a detailed of the code, please refer to the following anonymous URL, which has been provided due to page limitations: https://anonymous.4open.science/r/On-Truthful-Item-Acquiring-Mechanisms-for-Reward-Maximization-FD9C

