# OpenReview forum: "On Truthful Item Acquiring Mechanisms for Reward Maximization"
_ACM.org/TheWebConf/2024/Conference — TheWebConf24_

### Official Review · Reviewer_wus1 · 2023-11-21

**Novelty:** 6
**Technical Quality:** 5

**Review:**

The authors study an item acquiring problem between an owner and a collector. In particular, an owner of a set of items (and the one that knows their quality and holds this information private) wants to maximize the probability that the collector will acquire these items. The collector on the other hand, does not know the value of the items and they base their opinion on the evaluation of an expert (an evaluation that however has some noise and it is not always accurate). Their goal is to maximize the expected reward from the acquisitions. The paper is focused on truthful (from the side of the owner) mechanism design that produces good approximation ratios. The authors consider both the single item and the multi-item scenarios, and provide results regarding the performance of mechanisms, based on their menu size (the number of options that are given to the participants by the mechanism). They then complement their theoretical findings with some experimental results.

Strengths
1) The paper introduces and studies a very interesting and well-motivated problem. The general presentation is good, and the structure is nice. The authors consider both the single item and the multi-item version of the problem, and provide a good collection of results.
2) The paper is technically non-trivial, and as far as I checked, sound and correct.

Weaknesses
1) Parts of the paper are not that well-written, and in general you get the feeling that different people with completely different style of writing, wrote different sections of the paper. For example, the abstract is written in a very vague way, and the same goes for the contributions section as well (where I think that it is very difficult for someone that reads these parts to understand what the paper is about). The conclusions on the other hand, that summarizes the paper and the contributions, is written in a much better way.
2) Although I liked the results and I think that they are not trivial, I am not sure if any of them stands out as something very impressive. Of course, this is not necessarily a bad thing, but it kind of affects the significance of the contribution.

Overall, I would say that this is a nice paper, that studies a very interesting model and provides a set of results that are good to know. I feel though that probably it would benefit from a revision so that it becomes more consistent.

**Questions:**

None.

**Reviewer Confidence:**

3: The reviewer is confident but not certain that the evaluation is correct

**Scope:**

3: The work is somewhat relevant to the Web and to the track, and is of narrow interest to a sub-community

---

### Official Review · Reviewer_a1Hr · 2023-11-22

**Novelty:** 5
**Technical Quality:** 5

**Review:**

## After rebuttal

I thank the authors for answering my questions. I believe my concerns where adequately addressed, and as a result I am increasing my score.

## Contribution Summary:

The paper examines a paper where a collector wants to acquire items from an owner without monetary transactions. The items' quality is known to the owner but not to the collector. The collector knows a (possibly noisy) score by an impartial appraiser. The owner wants the item to be collected, and the collector wants to maximize her expected reward. The paper provides incentive compatible mechanisms for the item acquisition for one item, expands the discussion to multiple items and provides an experimental evalution for one and two items.

## Evaluation:

### Pros:

1. The paper works on an interesting problem, which is well connected to the conference and proposes an interesting new direction.
2. The writing of the paper is relatively clear, coherent, and the paper is mathematically rigorous in most cases.

### Cons:

1. The presented problem is analyzed with some relatively strong assumptions (see also Question 1).
2. The paper is vague in some cases (see questions 3,4,5,6 and 8 and related comments below).


Minor comments:

- Line 121: "Generally, a mechanism with a larger menu size has the potential to achieve higher efficiency by providing more choices to participants." I am not really sure that this is trivial knowledge. Can you provide some support on this statement?

- Line 187: Faltigs and Radaovic --> Faltings and Radanovic.

- Line 269: Can you please provide explicitly the definition of the collector's reward.

- Line 274: A more rigorous definition of monotonicity in the quality score would help here. For example, from the definition, it is unclear if the constant acquiring matrix $[x(v,s)=c]$ for all $v$ and $s$ is monotone.

* Line 307: the reference to the Appendix looks somewhat misplaced here.

* Line 508: "The stochastic matrix $R$ and the acquiring matrix $X$ by solving the linear program show below:". Please clarify if R was computed through the linear program or if only $X$ were the program's output.

* Line 884: missing space.

* Closely connected is the literature connected to strategic manipulations in peer review. You can see, e.g., the recent survey [Olckers, Walsh, 2022, Manipulation and Peer Mechanisms: A Survey]. (This is given as mere information to the authors in case they do not know about that part of the literature. This is not a suggestion for a citation!).

**Questions:**

1. One reservation I get about the paper's model is both $D$ and $R$ are public information, which are relatively strong assumptions. Can you provide some arguments on why these assumptions are needed?

2. This is closely connected to the previous question. Do you know how your results are affected when the distribution $D$ is unknown? E.g., if $D$ or $v$ are chosen adversarially, how the properties of the mechanism may change?

3.  I would find a case where the collector selected the quality threshold and not the nature more interesting, since the collector is the interested party. Can you support your choice? And do you believe the problem would be much different in this case?

4. Can you provide some intuition on the matrix $R$? This

5. Can you provide some intuition on the definition of total bias?

6. Can you explain briefly the connection of the row rank to the optimality of the mechanism?

7. Have you considered the simpler version of a one-menu, randomized mechanism? It seems a big step going from the score-only mechanism to the two-menu mechanism, and I wonder why you have not used the simpler step.

8. On Theorem 2.7, you show that the approximation ratio between pairs mechanisms is unbounded. I believe this has further implications which are not discussed. For example, what does this implies for the class of deterministic and incentive compatible mechanisms?

**Ethics Review Description:**

-

**Reviewer Confidence:**

3: The reviewer is confident but not certain that the evaluation is correct

**Scope:**

4: The work is relevant to the Web and to the track, and is of broad interest to the community

---

### Official Review · Reviewer_QsAb · 2023-11-26

**Novelty:** 5
**Technical Quality:** 5

**Review:**

This paper studies a problem where the goal is to design mechanisms that elicit truthful information from an owner about her collection of items with unknown quality. There are three parties in this problem: i) the owner: aims to maximize the probability of her items being sold, ii) the appraiser: an independent expert who assesses the items and gives them scores that are subject to random noise, and iii) the collector: relies on the owner's declaration and the appraiser's score to decide whether to acquire the item. The paper proposes several mechanisms for the single-item and multi-item settings and analyzes their properties and performance.

The problem studied in the paper is very relevant and has several applications in web-related domains, such as paper acceptance, app store review, and online auctions. The authors provide a comprehensive theoretical analysis of their proposed mechanisms including their incentive compatibility, monotonicity, optimality, and approximation ratio. Also, the numerical experiments are particularly interesting and they highlight the practicality of the proposed mechanisms for real-world applications. On the other hand, the paper's assumption that the appraiser is independent, non-strategic, and impartial is quite unrealistic. In many real-world situations, the appraiser also has conflicts of interest or biases that would impact their evaluation of the items. Also, the paper does not compare the proposed mechanisms with existing methods or baselines that deal with information asymmetry or elicitation problems, such as peer-prediction methods or scoring rules.

**Questions:**

- Is the assumption that the quality bar is item-independent in the multi-item setting reasonable? The utility of the collector for a collection of items might be a non-modular (for example, submodular) function of the items, and considering an item-independent quality bar does not seem realistic in many applications.
- How do you handle the case where the owner has multiple items with interdependent values or qualities? For example, if the owner has a collection of items that are more valuable as a whole than as separate pieces.
- How do you evaluate the robustness and scalability of your mechanisms in real-world scenarios, such as paper acceptance for conference proceedings or the app store review process? What are the main challenges and limitations of applying your mechanisms in these domains?
- Can you justify the assumption that the independent appraiser is non-strategic and impartial? How would your results change (or do they still hold) if the appraiser has some incentives or biases?
- Have you considered the possibility of collusion between some of these three parties (for example, the owner and the appraiser)?

**Reviewer Confidence:**

2: The reviewer is willing to defend the evaluation, but it is likely that the reviewer did not understand parts of the paper

**Scope:**

4: The work is relevant to the Web and to the track, and is of broad interest to the community

---

### Official Review · Reviewer_sXkX · 2023-11-29

**Novelty:** 6
**Technical Quality:** 4

**Review:**

This paper considers an interesting and novel problem of item-acquiring mechanism by eliciting the owner's information. I have concerns about this paper. One is that the utility of the owner does not depend on the value of items, which is not conventional in economics such as the bilateral trade model --- the true value of the item represents the opportunity cost of the sales. The other is that there is a major misinterpretation of the results on Su[35]: The Ranking Mechanism (RM) used in Theorem 3.1 does not satisfy assumptions in Su[35] --- the owner's utility (given by X) is not convex. Instead, the true reason that isotonic mechanism Su[35] does not apply in this setup is that the noise in appraisal is not necessarily exchangeable; this should be a more reasonable motivation for this setup. I would like to have the authors clarify on these issues.

That said, I do appreciate the cleanliness of the modeling and its well-presented results, so I would vote for borderline accepting this paper. I leave my other few questions and comments on some results of this paper to the section below.

**Questions:**

- The instance construction in Theorem 2.7 has X_{SOM} being 0 and thus the approximation ratio is unbounded. I wonder if we instead consider the additive approximation ratio, or require X_{SOM} to be non-zero, can we have better approximation guarantees?

- What if we enable more reporting power for the owners? That is, instead of always reporting a single value, the owner could commit to a signaling scheme that reports a distribution over the value conditioned on the true value. This setting makes sense if the owner (possibly as a large art broker) is also maximizing its expected utility over the distribution of items. This might help reduce the gap between the optimal mechanism and limited-menu mechanisms, based on recent results from information design literature.

**Reviewer Confidence:**

4: The reviewer is certain that the evaluation is correct and very familiar with the relevant literature

**Scope:**

4: The work is relevant to the Web and to the track, and is of broad interest to the community

---

### Decision · Program_Chairs · 2024-01-22

**Decision:**

Accept

**Comment:**

Most reviews are positive about the paper, and I will recommend weak acceptance.

 The writing of the paper is good. However, the technical contribution is marginal. The paper's problem can be seen as a special case of principal-agent games, and the main mechanism is based on standard linear programming. Here are are some additional minor points that the author should address carefully.
 - Reviewer sXkX pointed out that the paper improperly compared the ranking mechanism in Su[35] in their setting without acknowledging the violation of the ranking mechanism's original assumption.
 - Reviewer QsAb remarked that the paper needs to compare their results to existing mechanisms in peer-prediction methods or scoring rules. In particular, there may be interesting connections between their incentive-compatible mechanisms and surrogate proper scoring rule in [1].
 - Both Reviewer QsAb and a1Hr are concerned that the knowledge of and is relatively strong.


 [1] Liu, Yang, Juntao Wang, and Yiling Chen. "Surrogate scoring rules." ACM Transactions on Economics and Computation 10.3 (2023): 1-36.